# Efficient Long Context Fine-tuning with Chunk Flow

Xiulong Yuan [* 1]  Hongtao Xu [* 2 1 3]  Wenting Shen [1]  Ang Wang [1]  Xiafei Qiu [1]  Jie Zhang [1]  Yuqiong Liu [1]
Bowen Yu [1]  Junyang Lin [1]  Mingzhen Li [3]  Weile Jia [3]  Yong Li [1]  Wei Lin [1]

## Abstract

Long context fine-tuning of large language models(LLMs) involves training on datasets that are predominantly composed of short sequences and a small proportion of longer sequences. However, existing approaches overlook this long-tail distribution and employ training strategies designed specifically for long sequences. Moreover, these approaches also fail to address the challenges posed by variable sequence lengths during distributed training, such as load imbalance in data parallelism and severe pipeline bubbles in pipeline parallelism. These issues lead to suboptimal training performance and poor GPU resource utilization. To tackle these problems, we propose a chunk-centric training method named ChunkFlow. ChunkFlow reorganizes input sequences into uniformly sized chunks by consolidating short sequences and splitting longer ones. This approach achieves optimal computational efficiency and balance among training inputs. Additionally, Chunk-Flow incorporates a state-aware chunk scheduling mechanism to ensure that the peak memory usage during training is primarily determined by the chunk size rather than the maximum sequence length in the dataset. Integrating this scheduling mechanism with existing pipeline scheduling algorithms further enhances the performance of distributed training. Experimental results demonstrate that, compared with Megatron-LM, Chunk-Flow can be up to 4.53x faster in the long context fine-tuning of LLMs. Furthermore, we believe that ChunkFlow serves as an effective solution for a broader range of scenarios, such as long context continual pre-training, where datasets contain variable-length sequences.

---
[*]Equal contribution  [1]Alibaba Group  [2]School of Advanced Interdisciplinary Sciences, University of Chinese Academy of Sciences  [3]State Key Lab of Processors, Institute of Computing Technology, CAS. Correspondence to: Yong Li <jiufeng.ly@alibaba-inc.com>.

*Proceedings of the 42nd International Conference on Machine Learning*, Vancouver, Canada. PMLR 267, 2025. Copyright 2025 by the author(s).

## 1. Introduction

The ability of large language models (LLMs) to handle long contexts is crucial for tasks involving extensive inputs, such as comprehensive document analysis, extensive dialogue management, code generation from complex specifications, and detailed question answering over lengthy texts (Bai et al., 2024; OpenAI, 2024; Anthropic, 2025). Mainstream models like Llama can support context lengths of up to 128K tokens, while Google's Gemini can manage sequences up to 1M tokens(Gemini Team, 2024; Meta, 2024). However, given the significant costs associated with directly training models on long contexts, current practices typically involve an initial pre-training phase using shorter contexts (e.g., Llama with a context length of 8K), followed by a continual training stage to progressively extend their context lengths. Finally, Long context fine-tuning(Long SFT), which is the focus of this paper, is employed to further enhance the model's effectiveness in processing long contexts (Alibaba, 2025b; Meta, 2024).

To improve models' ability to handle long sequences without compromising their performance on short ones, Long SFT is typically conducted on datasets predominantly comprising short sequences with a small percentage of long ones (Xiong et al., 2024). For instance, Llama3 was fine-tuned using 99.89% short sequences (averaging under 1K tokens) and 0.11% long sequences (averaging around 37K tokens) (Meta, 2024). Current training methods, however, often prioritize long sequences at the expense of the unique characteristics of the long-tail distribution. This approach results in suboptimal training efficiency and inefficient GPU resource utilization (Zhao et al., 2024). The micro-batch size for each training step is usually determined by the memory requirements of long sequences, leading to significant under-utilization of GPU memory when processing mostly shorter sequences. Additionally, handling long sequences typically demands more GPUs than processing primarily short sequences. Existing methods allocate GPU resources based solely on the needs of long sequences, resulting in finer partitioning of computations. Consequently, this not only diminishes the training efficiency for short sequences but also exacerbates resource utilization issues. Moreover, variable sequence lengths in fine-tuning datasets also pose significant challenges in distributed training. For example, when

employing data parallelism, it causes load imbalance among data parallel ranks. When pipeline parallelism is used, it leads to severe pipeline bubbles. Although some studies aim to address the load imbalance issue (Bai et al., 2024), existing approaches fail to effectively tackle the pipeline bubble problem.

In this work, we introduce ChunkFlow to address the aforementioned issues. During each training step, given a sampled batch of sequences, ChunkFlow first reorganizes these sequences into new chunks. Specifically, it packs multiple short sequences into single chunks and divides longer sequences into several smaller ones, ensuring that the total sequence length of each chunk is approximately equal and does not exceed a predefined $ChunkSize$. The $ChunkSize$ is determined based on the training configuration and GPU memory constraints, aiming to strike an optimal balance between GPU computational efficiency and memory consumption. Next, ChunkFlow employs a state-aware chunk scheduling approach to manage these chunks during forward and backward passes. This scheduling method not only correctly handles computational dependencies among chunks from the same long sequence but also ensures nearly constant memory consumption mainly regulated by $ChunkSize$, regardless of the original sequence lengths. Furthermore, we integrate our state-aware scheduling algorithm with existing pipeline scheduling algorithms to significantly boost the performance of distributed training on sequences of variable lengths. In this paper, we implement a state-aware 1F1B (Shoeybi et al.) chunk scheduling method, reserving the integration with more advanced pipeline scheduling algorithms for future work. By employing this chunk-centered training method, ChunkFlow significantly improves end-to-end training performance for long context fine-tuning. Extensive experiments demonstrate that, compared with Megatron-LM, ChunkFlow achieves up to a 4.53x speedup in training performance for fine-tuning various sizes of Qwen2.5-series LLMs with different levels of context lengths. Moreover, we believe that ChunkFlow serves as an effective solution across a wide range of contexts, especially when dealing with training datasets containing sequences of variable lengths. This not only highlights its efficiency but also underscores its adaptability across diverse scenarios, showcasing ChunkFlow's versatility and broad applicability.

## 2. Preliminaries

In this section, we briefly walk through the preliminary literature related to this work.

### 2.1. Transformer and Causal Self-Attention

The Transformer architecture (Vaswani et al., 2017) has revolutionized deep learning and relies on self-attention

mechanisms to capture relationships between elements in a sequence. However, while self-attention is powerful, it inherently allows each token to attend to all other tokens in the sequence, including future ones. This characteristic is problematic for tasks like language modeling (OpenAI, 2024; Meta, 2024; Alibaba, 2025b), where predictions must be made sequentially—one token at a time—without access to future information. To address this limitation, causal self-attention introduces a causal mask, enforcing an auto-regressive property that ensures each token can only attend to previous tokens in the sequence. In this work, we leverage this causal property of LLMs to process long sequences in a chunk-by-chunk manner.

### 2.2. Sequence Packing

Sequence packing (Kosec et al., 2021) is a widely-used training technique for LLMs when dealing with datasets that contain variable-length sequences. This method involves concatenating multiple sequences within a batch into a single sequence, thereby eliminating the need for padding. As a result, it reduces redundant computations and memory usage. In this paper, we adopt sequence packing as the default approach.

### 2.3. Distributed Parallelism Strategies

In this section, we present commonly employed parallelization strategies for training LLMs. These strategies are frequently combined to achieve optimal training performance.

**Data Parallelism (DP)**: In data parallelism(Li et al., b; Rajbhandari et al.; Zhao et al.), the dataset is split into smaller subsets, and each subset is processed independently by a replica of the model running on different devices.

**Model Parallelism (MP)**: Model Parallelism is a critical technique for training large-scale deep learning models that exceed the memory capacity of a single device. It involves splitting a model into smaller components and distributing them across multiple devices, enabling the training of models with billions or even trillions of parameters. Model parallelism can be broadly categorized into two main approaches: Tensor Parallelism(TP)(Shoeybi et al.) and Pipeline Parallelism(PP)(Huang et al.; Narayanan et al., 2021). Tensor Parallelism(TP) divides individual layers or operations within a model across devices, ensuring that each device handles a portion of the tensor operations. Pipeline Parallelism(PP), on the other hand, splits the model into sequential stages, where each stage consists of one or more layers and is assigned to a different device(Fan et al.; Qi et al.). During training, the input data is processed stage-by-stage, with intermediate activations passed between devices. However, PP introduces inefficiencies known as pipeline bubbles, which occur when some devices remain idle during certain computational phases(Qi et al.). These bubbles reduce hard-

ware utilization and hinder training efficiency, making it essential to minimize pipeline bubbles for achieving high training performance and efficient utilization of distributed resources.

**Sequence Parallelism (SP)**: Sequence Parallelism(Li et al., a; Korthikanti et al.; Liu et al.; Jacobs et al.) is a technique used to distribute the computation of long sequences across multiple devices in LLMs. Instead of processing an entire sequence on a single device, the sequence is divided into smaller chunks, and each chunk is processed independently on different devices. This approach reduces memory usage per device and enables efficient training of models with long input sequences.

**Token-Level Pipeline Parallelism**: Token-Level Pipeline Parallelism(Li et al., c; Ao et al., 2025; Ma et al.) is a novel approach to parallelizing the computation of LLMs by exploiting the unique properties of causal attention . In this method, the input sequence is divided into smaller chunks at the token level, and these chunks are processed sequentially across multiple devices in a pipeline fashion. The key innovation lies in leveraging the causal masking property of causal attention, which ensures that each token only depends on its preceding tokens and not on future ones. This allows different pipeline stages to process distinct parts of the sequence independently, without violating the auto-regressive nature of the model. By assigning different token chunks to different pipeline stages, token-level pipeline parallelism reduces the memory burden on individual devices and improves hardware utilization. Furthermore, it reduces idle time (pipeline bubbles) by carefully scheduling computations and overlapping communication with computation. Overall, this approach enables efficient training of long sequences while maintaining the sequential dependencies required by causal attention.

## 3. Observations

Before delving into the design of ChunkFlow, we first present three key observations that motivate our work. To better illustrate these points, we use the example of training the Qwen2.5-7B model on the popular dataset *LM-SysChat1M* using Megatron-LM. However, it is crucial to emphasize that these insights are not limited to this specific scenario; rather, they are applicable to all long-context fine-tuning tasks.

**Observation 1: Extremely Long-tailed Sequence Length Distribution Characteristics in Datasets**.

Table 1 shows sequences distribution statistics in *LM-SysChat1M* dataset. It can be observed that over 99% of the sequences in the dataset are shorter than 4K tokens, while the longest sequence extends to approximately 300K tokens. This pronounced disparity results in an extremely

long-tail distribution of sequence lengths, posing significant challenges for efficient training and resource utilization. Notably, this distribution pattern has also been observed by Meta(Meta, 2024) as well as in our in-house proprietary training dataset, which is specifically collected for fine-tuning LLMs with context length over 256K.

| Sequence Length | Proportion of Sequences |
|---|---|
| $< 1K$ | 90.499% |
| $< 4K$ | 99.539% |
| $< 8K$ | 99.908% |
| $< 32K$ | 99.987% |
| $< 128K$ | 99.996% |
| Longest | 303K |

*Table 1.* Sequence Length Distribution in LMSysChat1M

**Observation 2: Apply Training Strategies Tailored For Long Sequences Causes Severe GPU Resource Underutilization and Poor Fine-tuning Efficiency**. When fine-tuning the Qwen2.5-7B model on the LMSysChat1M dataset with a context length of 32K (excluding sequences longer than 32K), we use 4 GPUs and set the global batch size to 64. If we directly apply a training strategy designed for long sequences, the micro-batch size has to be set to 1, and gradient accumulation over 64 micro-steps is required to prevent Out-Of-Memory (OOM) errors when processing sequences of 32K length. However, nearly 90% of the sequences in the dataset are shorter than 1K tokens. Given that memory consumption is proportional to sequence length, this approach results in significant underutilization of GPU resources during most micro-steps. Figure 1 depicts the memory usage across a randomly selected set of 1000 consecutive training micro-steps. The results reveal that while peak memory usage can soar up to 75GB, a staggering 97.7% of these micro-steps consume less than 45GB of memory. This clearly indicates suboptimal utilization of GPU resources.

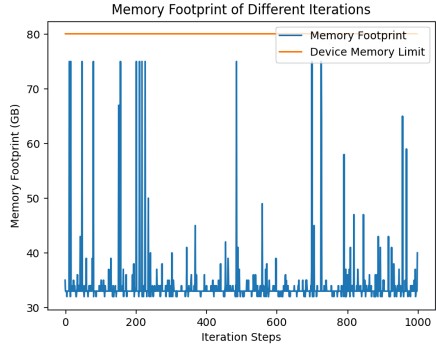

*Figure 1.* Memory Footprints in Different Iterations

Worse still, if we fine-tune the model with a 256K context

length (excluding sequences longer than 256K), we must allocate 16 GPUs to avoid OOM errors when handling sequences over 32K, even though these long sequences make up only 0.013% of the dataset. This approach requires distributing computations across 16 GPUs, significantly degrading the training performance for sequences shorter than 32K. Since sequences under 32K tokens dominate the dataset, using 16 GPUs not only reduces overall end-to-end training efficiency but also worsens resource utilization. Experiments show that partitioning computations across 16 GPUs instead of 4 GPUs results in a roughly 65% drop in training performance for sequences under 32K tokens.

**Observation 3: Variable Sequence Length Cause Severe Pipeline Bubbles**. The variation in sequence lengths within fine-tuning datasets also poses significant challenges for efficient distributed training, such as load imbalance among data parallel ranks in data parallelism and severe pipeline bubbles in pipeline parallelism. In this section, we primarily focus on the pipeline bubble problem.

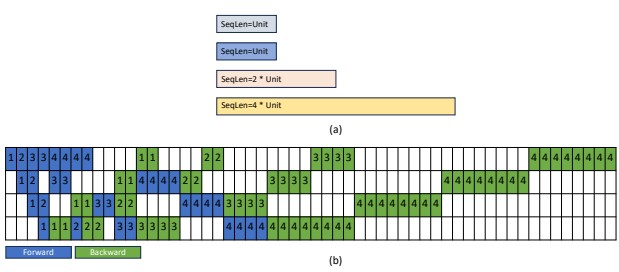

*Figure 2.* (a) 4 Sequences with Different Lengths;(b) Standard 1F1B Scheduling Result with $Pipeline\_Parallel\_World\_Size = 4$

We use the sequences in Figure 2(a) to explain pipeline parallel execution. The batch contains four sequences: two with $Unit$ tokens and two with $2 * Unit$ and $4 * Unit$ tokens, respectively. In addition to the commonly adopted assumption that the backward pass takes twice as long as the forward pass (Shoeybi et al.; Qi et al.), we introduce another reasonable assumption: the execution time of different sequences is proportional to their lengths. We also employ the bubble ratio, calculated according to Equation 1, to quantify the proportion of GPU time wasted by pipeline bubbles during the entire batch execution.

$$\text{Bubble Ratio} = \frac{\text{Total Bubble Time}}{\text{Total Execution Time}} \quad (1)$$

Figure 2(b) illustrates the scheduling results achieved by directly applying the standard 1F1B method to these four sequences with a $Pipeline\_Parallel\_World\_Size$ of 4. It can be observed that bubbles account for 57.14% of the total execution time. However, theoretically, when scheduling four sequences of equal length under this configuration, the bubble ratio should be 42.8%(Qi et al.; Narayanan et al., 2021). This discrepancy highlights that variable sequence lengths significantly exacerbate the pipeline bubble problem.

## 4. ChunkFlow

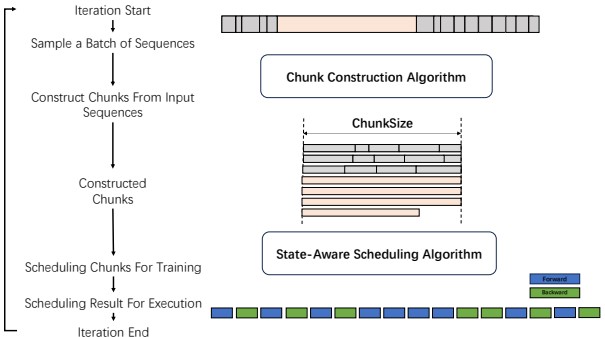

*Figure 3.* Overall Workflow in ChunkFlow

We illustrate the overall workflow of ChunkFlow in Figure 3. At each training step, given a sampled batch of sequences, ChunkFlow first reorganizes these sequences into a new set of chunks using a heuristic algorithm. This process involves merging multiple short sequences into single chunks and splitting longer sequences into several chunks as needed. Subsequently, a state-aware scheduling algorithm is employed to schedule these chunks for forward and backward passes. This scheduling approach not only properly manages computational dependencies between chunks from the same long sequence but also ensures nearly constant peak memory usage dictated by $K * ChunkSize$. Here, $ChunkSize$ is the length limit of the constructed chunks, while $K$ specifies how many chunks' activations should be saved by the scheduler. Additionally, we integrate ChunkFlow's scheduling algorithm with existing pipeline scheduling methods, such as 1F1B, and implement a state-aware 1F1B chunk scheduling method. This method can effectively enhance distributed training over variable-length sequences. We plan to incorporate ChunkFlow's idea into more advanced pipeline scheduling algorithms in future work. What's more, it should be noted that gradients from each chunk are accumulated to ensure mathematical equivalence with existing training methods.

### 4.1. Chunk Construction

For a given batch of sequences, we employ the heuristic algorithm described in Algorithm 1 to reorganize them into a new set of chunks. Sequences that exceed the $ChunkSize$ are divided into multiple chunks. For the remaining sequences shorter than the $ChunkSize$, the algorithm treats chunk construction as a bin packing problem with two con-

**Algorithm 1** ChunkConstructionAlgorithm

1: Given $ChunkSize, List[sequence]$.
2: Return $ResultChunks : List[Chunk]$ as result.
3: $LongSequences \leftarrow$ Select sequences longer than $ChunkSize$.
4: $ShortSequences \leftarrow$ Select sequences shorter than $ChunkSize$.
5: **for** $Sequence \in LongSequences$ **do**
6:     Divide $Sequence$ by $ChunkSize$ into multiple chunks and append chunks them to $ResultChunks$
7: **end for**
8: **for** $BinCnt = 1, \ldots, size\_of(ShortSequences)$ **do**
9:     $ResultBins \leftarrow$ Try binpacking $ShortSequences$ into $BinCnt$ bins with $ChunkSize$ as bin's max weight limit. If failed, continue trying for next $BinCnt$, otherwise, we take the binpacked result
10: **end for**
11: **for** $Bin \in ResultBins$ **do**
12:     Pack sequences in $Bin$ into a single chunk and add it to $ResultChunks$
13: **end for**

straints: the number of bins and the maximum weight limit per bin. The algorithm prioritizes minimizing the number of bins to maximize GPU computation efficiency.

Figure 4 illustrates an example of the chunk construction result from a batch of 16 input sequences. It shows that Sequence 6 is split into four chunks (Chunk 4 to Chunk 7), while the other shorter sequences are grouped into three chunks (Chunk 1 to Chunk 3).

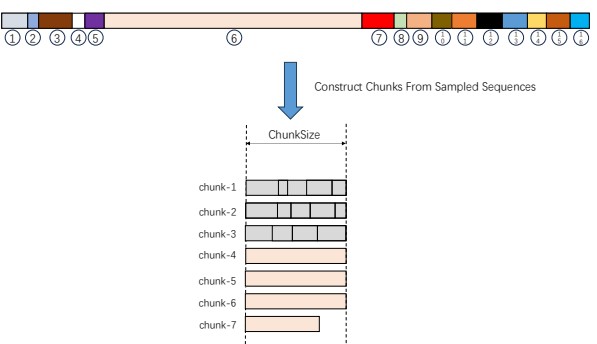

*Figure 4.* Chunk Construction Result from a Batch of 16 Sequences

## 4.2. State-Aware Chunk Scheduling

Given the chunks generated by Algorithm 1, we can observe two types of chunks: standalone chunks and dependent chunks. Standalone chunks contain complete sequences and can be processed independently. In contrast, dependent chunks contain segments of original long sequences and

thus rely on other chunks' states (primarily key/value tensors and their corresponding gradients in the causal attention modules) from the same long sequence to ensure correct computation. For example, in Figure 4, Chunk 1, Chunk 2, and Chunk 3 are standalone chunks, whereas Chunk 4 to Chunk 7 are dependent chunks. During training, standalone chunks do not require special scheduling and naturally consume memory according to the $ChunkSize$. However, dependent chunks must be carefully scheduled to ensure that peak memory usage remains within the limits defined by the $ChunkSize$. Given chunks extracted from the same long sequence (indexed from 1 to N), these chunks need to be processed in ascending order for forward passes and in descending order for backward passes. This is because, during forward propagation, subsequent chunks depend on the key/value tensors in the causal attention modules of preceding chunks, while during backward propagation, preceding chunks rely on the gradients of the key/value tensors from subsequent chunks. A naive scheduling approach would cause memory consumption to scale linearly with the length of the original sequence(Li et al., c; Ao et al., 2025).

To address this challenge, ChunkFlow introduces a state-aware chunk scheduling method detailed in Algorithm 2. Given a list of dependent chunks (where standalone chunks can be considered a special case with a list size of 1) and a parameter K, our scheduling approach ensures that memory consumption scales with $K * ChunkSize$ (with $K$ defaulting to 1) instead of the original sequence length. For scenarios where $N = sizeof(DependentChunks)$ and $N > K$, the forward passes of the first $(N - K)$ chunks are executed twice. During the initial forward pass, the activations of these chunks are discarded, while the causal attention modules' key/value tensors are stored as state for subsequent reuse during the second forward pass. Applying this chunk scheduling algorithm to all dependent chunk groups yields the final chunk execution order result. Figure 5 illustrates the scheduling results for the chunks constructed in Figure 4, with $K$ values set to 1 and 2, respectively. We observe that when K=1, Chunk 3 is executed twice, and at any given time during the entire execution, at most one chunk's activation is stored. However, when K=2, the activations of two chunks are retained, which results in higher memory consumption but also leads to improved end-to-end performance. It is worth noting that since Group-Query Attention(GQA)(Ainslie et al., 2023) has been widely adopted in mainstream large language models (LLMs) such as Llama, Qwen, and Gemini, the storage of the corresponding attention's key/value tensors and their gradients does not impose a significant memory overhead.

In summary, this scheduling method ensures a predictable memory usage pattern irrespective of the sequence length, while also achieving optimal computational efficiency.

**Algorithm 2** ChunkSchedulingAlgorithm

1: Given $DependentChunks = List[Chunk], K$.
2: $StateStore$ for sharing states across $Chunk$'s execution
3: $LossList$ for saving losses for each $Chunk$'s execution
4: **if** $size\_of(DependentChunks) <= K$ **then**
5:     **for** $Chunk \in DependentChunks$ **do**
6:         $Loss = model(Chunk, StateStore)$
7:         Append $Loss$ to $LossList$
8:     **end for**
9:     **for** $Loss \in reversed(LossList)$ **do**
10:         $backward\_with\_gradient\_accumulation(Loss)$
11:     **end for**
12: **else**
13:     **for** $Chunk \in DependentChunks$ **do**
14:         $Loss = model(Chunk, StateStore)$
15:         **if** $Chunk.Idx >= K$ **then**
16:             Append $Loss$ to $LossList$
17:         **else**
18:             Discard activations for $Chunk$
19:         **end if**
20:     **end for**
21:     **for** $Loss \in reversed(LossList)$ **do**
22:         $backward\_with\_gradient\_accumulation(Loss)$
23:     **end for**
24:     **for** $Chunk \in DependentChunks$ **do**
25:         **if** $Chunk.Idx < K$ **then**
26:             $Loss = model(Chunk, StateStore)$
27:             $backward\_with\_gradient\_accumulation(loss)$
28:         **end if**
29:     **end for**
30: **end if**

### 4.3. Incorporation with Pipeline Parallism: State-Aware 1F1B Algorithm

As mentioned in Section 3, directly applying the standard pipeline scheduling method to variable-length sequences can result in a relatively high bubble ratio, indicating wasted GPU idle time and poor end-to-end efficiency for distributed training. We incorporate ChunkFlow's idea with pipeline parallelism to solve this problem. Specifically, in this paper, we combine our state-aware scheduling mechanism with the standard 1F1B scheduling algorithm and implement a state-aware 1F1B scheduling method for efficient pipeline parallel execution over variable-length sequences.

For example, given the sequences in Figure 2, our state-aware 1F1B method operates as illustrated in Figure 6. When setting $ChunkSize = 2$, we obtain 4 chunks. Compared to the baseline method described in Section 3, our new scheduling approach achieves a reduced bubble ratio of 54.1% and an approximately 8% improvement in efficiency with $K = 1$. By increasing $K$ to 2, we further decrease the

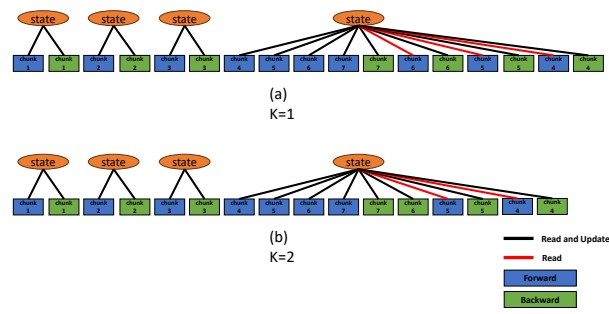

*Figure 5.* (a) Chunk Scheduling Result with $K = 1$;(b) Chunk Scheduling Result with $K = 2$

bubble ratio to 47.8%, resulting in a 12% enhancement in end-to-end efficiency.

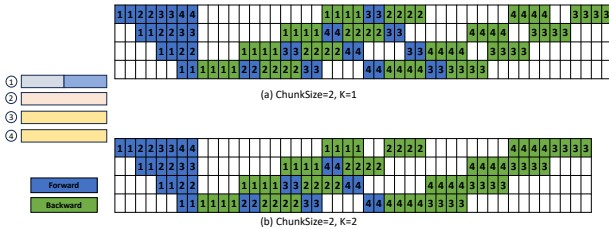

*Figure 6.* (a) State-Aware IF1B Scheduling Result with $ChunkSize = 2 * Unit$ and $K = 1$;(b) State-Aware IF1B Scheduling Result with $ChunkSize = 2 * Unit$ and $K = 2$

## 5. Determining $ChunkSize$ and $K$

The $ChunkSize$ and $K$ are primarily determined based on the training configuration and memory constraints, and they significantly impact end-to-end training performance. When pipeline parallelism is not utilized, $K$ should always be set to 1, and $ChunkSize$ should be maximized within memory constraints. This configuration aims to achieve maximum GPU utilization, thereby providing optimal training performance.

However, when pipeline parallelism is employed, we must carefully consider these two key parameters. A too large $ChunkSize$ results in fewer chunks, which can lead to more pipeline bubbles and degraded training performance. Conversely, a too small $ChunkSize$, while reducing pipeline bubbles, may fail to fully utilize the GPU's computational efficiency, leading to suboptimal overall performance. For example, setting $ChunkSize = 4 * Unit$ and $K = 1$ for the sequences in Figure 2 produces only 2 chunks. Scheduling them increases the bubble ratio to 60% and leads to a 15% performance degradation (as shown in Figure 7) compared with directly scheduling these four sequences using the standard 1F1B method. For a given training configuration, we employ a grid search approach

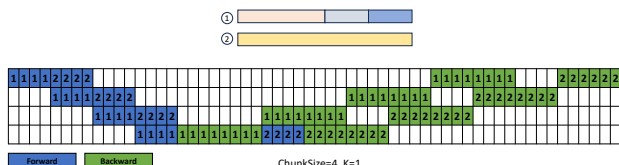

Figure 7. Unsuitable $ChunkSize$ and $K$ Leads to Performance Degradation

| Model | 32K | 256K |
|---|---|---|
| 7B | $< 4, 4, 1, selective >$ | $< 4, 4, 4, full >$ |
| 14B | $< 4, 4, 4, selective >$ | $< 4, 4, 4, full >$ |
| 32B | $< 4, 4, 4, selective >$ | $< 4, 4, 4, full >$ |
| 72B | $< 8, 8, 4, selective >$ | $< 8, 8, 4, selective >$ |

Table 3. Parallel Strategies for Training Different Models with Different Context Length, Formatted in $< TP, SP, PP, Recompute\ Granularity >$

| Model Size | 32K | 256K |
|---|---|---|
| 7B | $(32K, 1)$ | $(8K, 16)$ |
| 14B | $(8K, 8)$ | $(8K, 8)$ |
| 32B | $(8K, 6)$ | $(8K, 6)$ |
| 72B | $(8K, 16)$ | $(8K, 16)$ |

Table 4. Parameter Setting in ChunkFlow, Formatted in $(ChunkSize, K)$

to tune the parameters $ChunkSize$ and $K$. The additional overhead introduced by this process accounts for less than 0.5% of the total training time in our practice, and the resulting best combination is selected to maximize performance.

## 6. Evaluation

### 6.1. Evaluation Setup

We evaluate ChunkFlow on Qwen2.5-series LLMs and choose Megatron-LM as the baseline. This choice is due to the fact that Megatron-LM represents the state-of-the-art in LLM training, and ChunkFlow is built on top of it. Regarding the training dataset, we construct an evaluation dataset that better aligns with the real-world distribution characteristics for long context fine-tuning based on the methods provided in (Fu et al., 2024; ChatGLM, 2025). The sequence length distribution in the evaluation dataset is shown in Table 2. It can be observed that this evaluation dataset shares similar distribution characteristics with LMSysChat1M (Zheng et al., 2023), but has a slightly higher proportion of sequences longer than 32K and those shorter than 1K. All experiments are conducted using Alibaba Cloud ml.gu7ef.8xlarge-gu100 instances (Alibaba, 2025a), with a global batch size of 256 and a micro-batch size of 1.

| Sequence Length | Proportion of Sequences |
|---|---|
| $< 1K$ | 98.17% |
| $< 4K$ | 99.72% |
| $< 8K$ | 99.83% |
| $< 32K$ | 99.92% |
| $< 128K$ | 99.98% |
| Longest | 256K |

Table 2. Sequence Length Distribution in Evaluation Dataset

### 6.2. End-to-end Evaluation

We first evaluate the end-to-end training efficiency of using ChunkFlow for fine-tuning Qwen2.5 models listed in Table 3 on the evaluation dataset with context lengths of 32K and 256K, respectively. For each experiment, we exclude sequences exceeding the context length in the dataset and compare the performance with Megatron-LM. The configurations used for training different-sized models in Megatron-LM with various context lengths are shown in Table 3. These configurations achieve the best performance in Megatron-LM while ensuring no OOM errors occur.

ChunkFlow adopts the same parallel strategies as the baseline (i.e., the combination of $< TP, SP, PP >$) but incorporates the selective recomputation strategy across all experiments. This is because ChunkFlow's memory consumption is primarily determined by the $ChunkSize$ rather than the length of the longest sequence, allowing it to avoid OOM issues without requiring full recomputation strategy. Additionally, we list the best-performance configurations for ChunkFlow, obtained through a grid search over $ChunkSize$ and $K$, in Table 4.

Figure 8 illustrates the performance results of ChunkFlow and Megatron-LM, with performance measured by average iteration time. To facilitate a clearer comparison, we normalize Megatron-LM's results relative to those of ChunkFlow. The results indicate that ChunkFlow can accelerate long context fine-tuning by up to 4.53x. This significant improvement in ChunkFlow can be attributed to two key design innovations. First, ChunkFlow consolidates short sequences into single chunks, which greatly enhances computational efficiency. Second, it's state-aware 1F1B scheduling mechanism reduces pipeline bubbles and thus further boosts overall performance. Additionally, unlike Megatron-LM, ChunkFlow avoids memory bottlenecks, eliminating the need for full recomputation when fine-tuning models with 256K context lengths, such as 7B, 14B, and 32B models. Notably, even when using the same recomputation strategy, experiments on fine-tuning the 72B model demonstrate that ChunkFlow still achieves substantial performance gains, underscoring its superior efficiency and scalability.

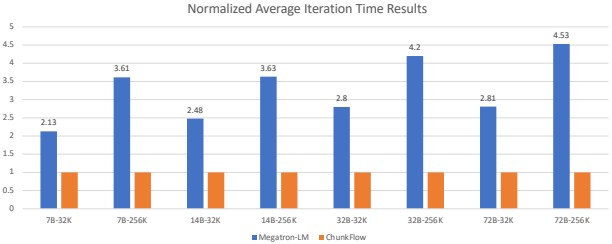

*Figure 8.* Normalized End-to-End Training Performance Results

### 6.3. Case Study

In this section, we conduct experiments to explore the memory consumption characteristic of ChunkFlow as well as the performance impact of the $ChunkSize$ and $K$.

#### 6.3.1. MEMORY CONSUMPTION CHARACTERISTIC IN CHUNKFLOW

Table 5 demonstrates the peak memory usage when fine-tuning the 7B model with different context lengths and varying $ChunkSize$. All experiments share the same training configuration of $< 4, 4, 1, selective >$, and $K$ is set to 1. The results indicate that ChunkFlow's memory consumption is primarily determined by the $ChunkSize$ regardless of the maximum sequence length in datasets. However, it can also be observed that training with a 256K context length consumes slightly more memory compared with training with a 32K context length using the same $ChunkSize$. This is because, in our current implementation, we directly save all key/value tensors in memory without further offloading optimizations. We leave this optimization for future work.

| Context Length | $ChunkSize$ | Peak Memory |
|----------------|-------------|-------------|
| 32K            | $2K$        | 41.6 GiB    |
| 256K           | $2K$        | 45.6 GiB    |
| 32K            | $4K$        | 47.5 GiB    |
| 256K           | $4K$        | 50.8 GiB    |
| 32K            | $8K$        | 59.3 GiB    |
| 256K           | $8K$        | 63.8 GiB    |

*Table 5.* Memory Consumption Characteristic in ChunkFlow

#### 6.3.2. PERFORMANCE IMPACT OF $ChunkSize$ AND $K$

We also investigate the performance impacts of $ChunkSize$ and $K$ by fine-tuning the 7B model with a context length of 256K and various combinations of these two parameters. All experiments are conducted using the $< 4, 4, 4, selective >$ training strategy, and training performance is measured in terms of average iteration time. We maintain $ChunkSize *$ $K$ constant across different settings to ensure that all experiments save the same total amount of chunk activations

for a dependent chunk group. The results in Table 6 indicate that both $ChunkSize$ and $K$ significantly influence overall performance. The $(8K, 4)$ configuration achieves optimal performance. In contrast, the $(2K, 16)$ configuration leads to suboptimal performance since it yields chunks that are too small to fully exploit the GPU's computational efficiency. Additionally, the $(32K, 1)$ configuration results in fewer chunks, leading to increased pipeline bubbles and consequently reducing overall performance.

| $(ChunkSize, K)$ | Training Performance(ms) |
|------------------|--------------------------|
| $(2K, 16)$       | 29810                    |
| $(8K, 4)$        | 23774                    |
| $(32K, 1)$       | 28942                    |

*Table 6.* Impacts of $ChunkSize$ and $K$ on Training Efficiency

## 7. Related Works

A significant portion of research in the field of long context fine-tuning primarily focuses on enhancing model performance by meticulously constructing datasets (Meta, 2024; Alibaba, 2025b; Bai et al., 2024; Zhao et al., 2024). However, the training efficiency of this task has not been thoroughly explored. As the importance of long context fine-tuning grows, there is an increasing emphasis on improving end-to-end training performance by leveraging the unique characteristics of this workload, particularly the variability in dataset lengths. Sequence packing is proposed as a method to eliminate padding when processing batches with varying sample lengths, thereby reducing unnecessary memory usage and computational demands (Kosec et al., 2021). Additionally, smart batching (Bai et al., 2024), which sorts mini-batches during training steps, is employed to achieve better load balancing among data parallel ranks. Furthermore, several studies that do not directly address the issue of long context fine-tuning have contributed solutions to related challenges within this domain. HotSPa (Ge et al., 2024) introduces multiple sequence parallel strategies tailored for varying sequence lengths in the training datasets. There is also a line of research, such as LoRA(Hu et al., 2022), that focuses on parameter-efficient fine-tuning to improve training efficiency. These approaches are orthogonal to ChunkFlow and can be seamlessly integrated with it.

## 8. Conclusion

In this work, we propose a chunk-centric training method called ChunkFlow to effectively address the challenges posed by the unique distribution characteristics of long-context fine-tuning datasets for LLMs. ChunkFlow reorganizes input sequences into uniformly sized chunks by consolidating short sequences and splitting long ones. It also introduces a novel state-aware chunk scheduling algo-

rithm to manage these chunks for forwards and backwards. This scheduling algorithm ensures nearly constant memory consumption, which is governed by the chunk size. Additionally, we integrate ChunkFlow's approach with existing pipeline scheduling methods and implement a state-aware 1F1B scheduling technique , further enhancing distributed training performance on variable-length sequences. Evaluation results demonstrate that ChunkFlow achieves up to a 4.53x speedup in performance compared to the state-of-the-art training system, Megatron-LM. Furthermore, we believe that ChunkFlow can serve as an effective solution for a broader range of scenarios involving the training of LLMs on variable-length sequences.

## Impact Statement

This paper presents work whose goal is to advance the system efficiency of long-context fine-tuning. There are many potential societal consequences of our work, none which we feel must be specifically highlighted here.

## Acknowledgements

This work is supported by the following funding: National Science Foundation of China (92270206, 62372435), China National Postdoctoral Program for Innovative Talents (BX20240383) and Beijing Natural Science Foundation (4254087).

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
