# OpenReview forum: "Efficient Long Context Fine-tuning with Chunk Flow"
_ICML.cc/2025/Conference — ICML 2025 poster_

### Official Review · Reviewer_dcDa · 2025-03-11

**Overall Recommendation:** 3

**Summary:**

This paper introduces ChunkFlow, an LLM training (fine-tuning) method that aims to improve the computational as well as memory efficiency of long-context training / fine-tuning. The authors start from three empirical observations in long-context fine-tuning, point out existing efficiency bottlenecks, and design ChunkFlow based on them. Given a batch of input sequence, ChunkFlow reorganizes it to a new set of chunks based on heuristics, so that the size of each chunk does not exceed a pre-defined chink size. The method also incorporates state-aware chunk scheduling and state-aware 1F1B for better training efficiency. Evaluations show that as compared to Megatron-LM, ChunkFlow accelerates long-context fine-tuning by up to 4.53x.

**Claims And Evidence:**

The claims made by the authors in this paper are supported by citations or experiment results.

**Essential References Not Discussed:**

N/A

**Experimental Designs Or Analyses:**

No particular flaw in experiment design. See issues/concerns in "Questions For Authors".

**Methods And Evaluation Criteria:**

No particular flaw in evaluation design. See issues/concerns in "Questions For Authors".

**Other Comments Or Suggestions:**

N/A

**Other Strengths And Weaknesses:**

Issues not mentioned in prior sections:
- Missing an impact statement.

**Questions For Authors:**

Thank you for submitting this paper to NeurIPS. This is a well-motivated paper, drawing inspirations from empirical observations and presenting a concrete solution that is well-evaluated.

Below are a few questions and concerns:
- The figures do a good job in illustrating different parts of the system (esp. reorganizing the chunks) but really need to be fixed. Fonts are too small to be read clearly (esp. figures 2, 3, 5, 6, 7).
- You mention that you use grid search to find best K and chunk size --- I am curious what the cost is for the grid search process, before fine-tuning begins. And, in particular, how that cost compares to the training cost itself, e.g. the cost in terms of latency in table 6.
- How does ChunkFlow generalize to pre-training? I understand that pre-training experiments might be too hard to implement, but I am curious how and why many of the design choices/intuitions in ChunkFlow could help improve the computational/memory efficiency of pre-training.

**Relation To Broader Scientific Literature:**

This paper lies within the broader area of LLM fine-tuning systems. In particular, it addresses the computational and memory efficiency of long-context fine-tuning (a special case of LLM fine-tuning).

**Theoretical Claims:**

N/A

---

> ### Author Rebuttal · Authors · 2025-03-31
>
> We are sincerely grateful to Reviewer **dcDa** for devoting time to review our work and providing invaluable feedback. Regarding the clarity of figures and the impact statement, we will, in strict compliance with Reviewer dcDa’s suggestions, **incorporate an impact statement and refine all figures—with particular focus on Figures 2, 3, 5, 6, and 7—in the final version of our release**. Moreover, we offer a comprehensive response below to other concerns raised by Reviewer dcDa.
>
> ***(Q1) The cost of grid search compares to training cost itself***
>
> We sincerely appreciate reviewer dcDa's meticulous review of our paper and point out a key implementation detail regrading to grid search and allows us to furtherly clarify the whole training process of ChunkFlow. When compared to the remarkable benefits of identifying the optimal `(ChunkSize, K)`, the overhead of grid search is almost negligible.
> Below, we use the grid search process presented in Table 6 to illustrate this. In our research on the 72B model (relevant data can be found in Table 6), we evaluated several configurations, namely `[(32K, 1), (16K, 2), (8K, 4)]` . To accurately estimate training performance, each `(ChunkSize, K)` combination had to undergo 10 training steps. The entire grid search process accumulated 15 min. Significantly, training with a full 256K-context using ChunkFlow takes approximately multiple days or even weeks(~70 hours in our case). **Evidently, the overhead of grid search accounted for less than 0.3% of the total training time**.
>
> From a cost-benefit perspective, the optimal configuration `(8K, 4)` achieved a **1.21x** speed-up compared to the worst-performing configuration(i.e.`(2K, 16)` ). As a result, approximately **12 hours of training time are saved** over the entire training process in our case.
> Moreover, these validated optimal parameters can be reused across repeated training runs, such as training new 72B model versions with new algorithms or training on updated datasets. This effectively amortizes the one-time cost of grid search.
>
> ***(Q2): How and why many of the design choices/intuitions in ChunkFlow could help improve the computational/memory efficiency of pre-training***
>
> Although ChunkFlow is mainly designed to boost training efficiency for LLM's post-training phase like SFT and RLHF which involve training on variable-length sequence datasets. It's design can also be applied to other training stages like pretraining and long context continual pretraining.
>
> As mentioned in Meta's tech report[1], there are three main stages in developing a new LLM model: pretraining, long context continual pretraining (for context length extension), and post-training (for human preference alignment). In Section 1(Introduction), we claim that ChunkFlow can enhance long context continual pretraining efficiency because this phase also involves training on datasets with variable-length sequences[1] and exhibits efficiency challenges similar to those in the SFT stage (e.g., pipeline bubbles and resource underutilization). **Consequently, ChunkFlow’s design naturally improves computational and memory efficiency in long context continual pretraining scenario**.
>
> Regarding pretraining, even though it typically operates on uniformly sized sequences, splitting sequences into chunks and utilizing ChunkFlow's state-ware scheduler to process them sequentially can still reduce pipeline bubbles(due to increased number of micro-batches)and alleviate memory pressure. **This suggests that ChunkFlow’s chunk-centric approach retains value even in scenarios with uniform sequence lengths**.
>
> [1] Meta(2024). The Llama 3 Herd of Models. ArXiv, abs/2407.21783.

---

> > ### Comment · Reviewer_dcDa · 2025-04-08
> >
> > I appreciate the authors for their detailed and helpful response! I have also read other reviews in detail. At this point, I don't have more questions related to design and evaluation, and I decide to maintain my score.

---

> > > ### Author Response · Authors · 2025-04-08
> > >
> > > We sincerely appreciate the reviewer dcDa's time and valuable feedback on our work, we will keep moving to make Chunkflow better.

---

### Official Review · Reviewer_xBtd · 2025-03-13

**Overall Recommendation:** 4

**Summary:**

This paper propose Chunk Flow, a novel chunking and scheduling method for pipeline parallel long sequence training. It first discussed the long-tail phenomena of long-context LLMs training, and the potential issues result from it, such as underutilize of GPU memory and pipeline bubbles. Then the authors propose to chunk the longer sequence to a max chunk size, with a scheduler handling the defendency of the causal mask of long sequence. The author compares chunk flow against the state of the art pre-training framework megatron and find chunk flow well maintained the peak memory across various length long sequence during the training process, while achieve a 4x acceleration compared to megatron.

**Claims And Evidence:**

-

**Essential References Not Discussed:**

-

**Experimental Designs Or Analyses:**

-

**Methods And Evaluation Criteria:**

The strength of this paper:
1. Addressing a critical real world problem. Training long sequence model is very expensive and many existing frameworks have very limited optimization on long sequence pre-training. Chunk Flow ensures a upper bound of the peak memory and reduce pipeline bubble which can help the scaling of sequence length and size of LLMs.

2. Good empirical results. The proposed method achieves good peak memory optimization, enables lower 60GB memory with 8k chunk size, which is quite impressive. The end2end speedup is 4x faster against megatron, which can significantly reduce the cost to train long sequence LLMs.

The limitation and questions for the authors:
1. When the N > K, ie, the longest sequence exceed the num of chunks, it did duplicate computations for the overlapping chunks, and it keeps dependent chunk on GPU, which can leads to significant overhead.

2. It is unclear that what components are actually retained for dependent sequence, especially within causal attention.

3. What is the training data has more longer sequence than shorter sequence, how much is the benefit to use chunk flow in this scenarios.

**Other Comments Or Suggestions:**

This is overall a solid study and have provide well explaination for the majority of the paper. Good experimental results. But the author should redo the figures as they are not quite readable.

**Other Strengths And Weaknesses:**

-

**Questions For Authors:**

-

**Relation To Broader Scientific Literature:**

-

**Theoretical Claims:**

-

---

> ### Author Rebuttal · Authors · 2025-03-31
>
> We thank **xBtd** for their very positive review of our work, noting that our paper is "a solid study and have provide well explaination for the majority of the paper." and emphasizing strongly the importance of the research problem. In accordance with reviewer xBtd’s suggestions, we will redo all figures to enhance their readability in the final release. Additionally, we provide detailed responses to reviewer xBtd’s other concerns.
>
> ***Q1: Overhead of keeping dependent chunks when N > K***
>
> We sincerely appreciate the reviewer’s attention to the issue of computational overhead when the number of dependent chunks (N) exceeds K. In ChunkFlow, when N>K, the forward passes of the first `N-K` chunks are executed twice. After the first-time forward passes of these `N-K` chunks, **we discard their activation values**. Instead, **we only retain the key/value tensors of their attention mechanisms as states for subsequent reuse**.
>
> Since Multi-Query Attention (MQA) and Group-Query Attention (GQA) are widely adopted in mainstream large language models (LLMs) like LLama, Qwen, and Gemini, **storing these attention key/value tensors does not impose a substantial burden**. As can be seen from Table 5 in the paper, compared to processing 32K-length sequences, processing 256K-length sequences only consumes approximately 4GB more memory(Still much less memory footprint compared to Megatron-LM baseline). This clearly demonstrates that retaining these states in memory does not pose a severe problem.
>
> When training models with longer contexts, such as 2M token context models, retaining these states could ultimately lead to excessive memory consumption. As noted in Section 6.3.1, we plan to optimize this memory consumption through carefully designed memory offloading strategies.
>
> ***Q2: Components retained for dependent sequences in causal attention***
>
> For dependent chunks (split from long sequences), ChunkFlow retains key/value (K/V) tensors and their gradients in causal attention layers. These components are critical for:
>
> 1. Forward Dependency : Subsequent chunks depend on K/V tensors from prior chunks to maintain causal masking (Section 4.2).
>
> 2. Backward Dependency : Gradients for K/V tensors in earlier chunks require activations from later chunks(Figure 5).
>
> ***Q3: Efficacy in datasets dominated by long sequences***
>
> Even when long sequences are more prevalent in datasets, ChunkFlow continues to demonstrate substantial advantages. However, the extent of performance improvement hinges on the distribution of data. For scenario analysis, we classify sequences as long if they exceed ChunkSize; otherwise, they are categorized as short. Whenever short sequences are present, we can consolidate them into a single chunk, significantly enhancing the utilization of GPU resources. As demonstrated in Figure 6, for long sequences, ChunkFlow splits them into uniformly sized chunks, effectively reducing pipeline bubbles. Subsequently, we carried out an experiment on the LongBench dataset [1]. The dataset exhibits a sequence-length distribution pattern consistent with what the reviewer envisioned.
>  LongBench dataset is used for bilingual, multitask, and comprehensive assessment of long context understanding capabilities of LLMs and the table below demonstrates sequence length distribution in the dataset. When setting ChunkSize=8K, we can see that more than 50% sequences are long sequences and ChunkFlow achieve **1.7X** speeup over Megatron-LM, showing its effectiveness in a broader scenario.  If we further filter the LongBench dataset by removing sequences with fewer than 8K tokens, we can obtain a new dataset that **solely consists of long sequences**. Thanks to the Chunking mechanism of ChunkFlow, which can reduce pipeline bubbles, ChunkFlow still achieves a **1.4X** speedup compared to Megatron-LM.
>
> | Sequence Length| Propotion Of Sequences |
> |---------|---------|
> | < 1K | 0.26% |
> | < 4K | 22.65% |
> | < 8K | 48.4% |
> | < 16K | 82.59% |
> | < 32K | 98.06% |
> | Longest | 64K |
>
> [1] Bai, Y., Lv, X., Zhang, J., Lyu, H., Tang, J., Huang, Z., Du, Z., Liu, X., Zeng, A., Hou, L., Dong, Y., Tang, J., & Li, J. (2023). LongBench: A Bilingual, Multitask Benchmark for Long Context Understanding. ArXiv, abs/2308.14508.

---

### Official Review · Reviewer_HVAi · 2025-03-14

**Overall Recommendation:** 3

**Summary:**

This paper introduces ChunkFlow, a novel method for efficient long-context fine-tuning of large language models (LLMs). ChunkFlow addresses the challenges of variable sequence lengths in training datasets by reorganizing input sequences into uniformly sized chunks, merging short sequences and splitting long ones. It employs a state-aware chunk scheduling mechanism to manage computational dependencies and ensure consistent memory usage, primarily determined by the chunk size. The method also reduces pipeline bubbles and improving distributed training efficiency.

**Claims And Evidence:**

The paper claims that ChunkFlow improves training efficiency for long-context fine-tuning by reorganizing sequences into uniform chunks and using state-aware scheduling. Evidence in the experiments in Section 6 includes the comparison to Megatron-LM, with Peak Memory and training performance.

**Essential References Not Discussed:**

The paper does not discuss other efficient training methods like LoRA, or other  long-sequence papers, like  RingAttention.

**Experimental Designs Or Analyses:**

Experiments involve fine-tuning Qwen2.5-series LLMs with varying context lengths, comparing ChunkFlow to Megatron-LM on metrics like memory consumption and training speed.

**Methods And Evaluation Criteria:**

ChunkFlow uses chunk construction and state-aware scheduling to manage variable-length sequences. Evaluation is based on training performance metrics like iteration time and memory usage.

**Other Comments Or Suggestions:**

N/A

**Other Strengths And Weaknesses:**

This paper did not compare with any other long-context models, on any long-context benchmark, like LongBench.

**Questions For Authors:**

How does ChunkFlow perform on datasets with even longer sequences (e.g., >1M tokens)?
Could the method be extended to non-causal models?

**Relation To Broader Scientific Literature:**

ChunkFlow builds on existing work in sequence packing and pipeline parallelism.

**Theoretical Claims:**

The paper argues that chunk-based training and state-aware scheduling can optimize GPU utilization and reduce pipeline bubbles in distributed training. This is the claim, but no so theoretical.

---

> ### Author Rebuttal · Authors · 2025-03-31
>
> We sincerely appreciate the reviewer **HVAi**'s  insightful feedback. Below we clarify the concerns regarding related work discussion and broader evaluations, with references to our methodology and results in the paper.
>
> ***(Q1): The paper does not discuss other efficient training methods like LoRA, or other long-sequence papers, like RingAttention.***
>
> We would like to clarify that ChunkFlow emerged in response to the practical need to accelerate the fine-tuning of LLMs on datasets with sequences of varying lengths. Within this context, we discuss several relavant directions in Section 2(Premilinaries) and Section 7(Related Works). We also give a brief introduction about long sequence training methods such as sequence parallelism and token-level pipeline parallelism in Section 2 and RingAttention is also referenced in this section.
>
> **It is worth to highlight that ChunkFlow is orthogonal to LoRA and Ring-attention**. We conduct a discussion here to provide further explanations.
>
> ● LoRA focuses on Parameter-Efficient Fine-Tuning by substantially reducing the number of training parameters. Conversely, ChunkFlow aims at enhancing the training efficiency of LLMs on variable-length datasets. Moreover, integrating ChunkFlow into the LoRA training process can effectively boost the training efficiency of LoRA.
>
> ● RingAttention is an effective approach for handling extremely long sequences. However, it does not address training issues such as low resource utilization and pipeline bubbles caused by the variability in sequence lengths. When performing long context finetuning at scale, ChunkFlow and the RingAttention can complement each other. The RingAttention mechanism offers a method for distributed attention computation. ChunkFlow, through its unified chunking and state-aware scheduling strategy, reduces pipeline bubbles and enchances computation efficiency (as depicted in Figure 6), thereby improving training performance. We will incorporate the above-mentioned suggestions into the final version.
>
> ***(Q2): This paper did not compare with any other long context models, on any long context benchmark, like LongBench.***
>
> We sincerely appreciate the valuable suggestions put forward by reviewer HVAi and we conduct more experiments using the ***Llama3-8B*** model on LongBench dataset. The results and analysis will also be reflected in our final version.
>
> It is worth to clarify that the LongBench dataset is primarily employed to evaluate the long context understanding capabilities of LLMs, rather than for long context fine-tuning.  The table below presents the distribution characteristics of sequence lengths in LongBench dataset and demonstrate significantly difference distribution with our previously mentioned SFT dataset, as well as Llama SFT dataset[1]. ***Those distinctions helps explain why LongBench wasn't initially included in our experiment datasets.***
>
> | Sequence Length | Propotion Of Sequences in LongBench| Propotion Of Sequences in Meta |
> |:-------|:--------:|-------:|
> | < 1K  | 0.26%    |  ~ 99% |
> | < 4K  | 22.65%   | - |
> | < 8K  | 48.4%| -  |
> | < 16K  | 82.59%   | -  |
> | < 32K  | 98.06%   | -  |
> | Longest| 64K   | 128K  |
>
> However, we agree that benchmarking on LongBench provides important validation of ChunkFlow's effectiveness, and we conduct the experiments using llama3-8B on longbench to further highlight our contributions. For ChunkFlow, we configure the `ChunkSize=16384, K=2`. Both experiments adopt the identical `<TP=2, SP=2, PP=4>` parallelization strategy. Under these circumstances, ChunkFlow demonstrates a **1.7x** speedup compared to Megatron-LM. Evidently, this showcases the superiority of our design.
>
> ***(Q3): How does ChunkFlow perform on datasets with even longer sequences (e.g., >1M tokens)? Could the method be extended to non-causal models?***
>
> As shown in Figure 8, ChunkFlow shows greater performance improvement when fine-tuning models with longer context lengths. As the fine-tuning context length increases, the sequence length distribution widens. Training strategies for the longest sequences perform more poorly for most short ones. This implies that for models with even longer contexts (e.g., >1M tokens), ChunkFlow can achieve far better performance than baselines.
>
> For applying ChunkFlow to non-causal model, it has two chunk-construction methods: short-sequence consolidation and long-sequence splitting. The former can be directly used in non-causal model training. The latter depends on causal-attention. So, when using ChunkFlow for non-causal models, setting ChunkSize to the dataset's maximum sequence length can effectively improve training efficiency.
>
> Due to limitations in computational resources and the timeline for this submission, we regret that we were unable to include these specific (>1M, speedups on LoRA and non-causal) experiments. We will incorporate the above-mentioned experiments in the final version.
>
> [1] Meta(2024). The Llama 3 Herd of Models. ArXiv, abs/2407.21783.

---

> > ### Comment · Reviewer_HVAi · 2025-04-02
> >
> > Thanks for the detailed reply. My concerns have been resolved. I increased the rate for this paper.

---

> > > ### Author Response · Authors · 2025-04-07
> > >
> > > Thank you for your thoughtful review and recognizing the importance of our work.

---

### Decision · Program_Chairs · 2025-05-01

**Decision:**

Accept (poster)

**Comment:**

The paper introduces ChunkFlow, a method designed to enhance the efficiency of fine-tuning LLMs on long context data by reorganizing input sequences into uniformly sized chunks and employing a state-aware chunk scheduling mechanism. This approach aims to address common issues in distributed training, such as load imbalance and pipeline bubbles, which can lead to suboptimal training performance and underutilized GPU resources. By ensuring optimal computational efficiency and balance, ChunkFlow claims to accelerate training performance by up to 4.53 times compared to the Megatron-LM baseline, with applications beyond fine-tuning, such as continual pre-training and pre-training stages.
It addresses a critical real-world problem in the training of LLMs, particularly in terms of memory and computational efficiency.
In the rebuttals the authors addressed the reviewers' concern well.